# Mechanisms and Physiological Roles of Polymorphisms in Gestational Diabetes Mellitus

**DOI:** 10.3390/ijms25042039

**Published:** 2024-02-07

**Authors:** Sarocha Suthon, Watip Tangjittipokin

**Affiliations:** 1Department of Immunology, Faculty of Medicine Siriraj Hospital, Mahidol University, Bangkok 10700, Thailand; sarocha.suo@mahidol.edu; 2Siriraj Center of Research Excellence for Diabetes and Obesity, Faculty of Medicine Siriraj Hospital, Mahidol University, Bangkok 10700, Thailand; 3Siriraj Center of Research Excellence Management, Faculty of Medicine Siriraj Hospital, Mahidol University, Bangkok 10700, Thailand

**Keywords:** SNPs, gestational diabetes mellitus, genetics, GWAS

## Abstract

Gestational diabetes mellitus (GDM) is a significant pregnancy complication linked to perinatal complications and an elevated risk of future metabolic disorders for both mothers and their children. GDM is diagnosed when women without prior diabetes develop chronic hyperglycemia due to β-cell dysfunction during gestation. Global research focuses on the association between GDM and single nucleotide polymorphisms (SNPs) and aims to enhance our understanding of GDM’s pathogenesis, predict its risk, and guide patient management. This review offers a summary of various SNPs linked to a heightened risk of GDM and explores their biological mechanisms within the tissues implicated in the development of the condition.

## 1. Introduction

Gestational diabetes mellitus (GDM) refers to diabetes diagnosed in the second or third trimester of pregnancy that was not present before gestation and carries risks for the mother, fetus, and newborn. GDM is often indicative of underlying β-cell dysfunction and increases the risk for later development of diabetes, which is often type 2 diabetes (T2D), in the mother after delivery [1]. 

The maternal body undergoes a sequence of physiological changes to accommodate the needs of the developing fetus, with insulin sensitivity representing a vital metabolic adaptation during this process. An upsurge in local and placental hormones during mid- to late gestation induces insulin resistance. GDM occurs when pancreatic β-cells cannot adequately respond to the heightened requirement for insulin secretion, leading to the onset of spontaneous hyperglycemia during pregnancy. Both the malfunction of β-cells and the resistance of body tissues to insulin are integral aspects of the pathophysiology of GDM [2].

GDM is a multifactorial disorder that is influenced by interactions between genetic and environmental factors. The interplay between these factors suggests the complexity of the mechanistic pathways underlying GDM. Genome-wide association studies (GWAS) have identified specific genetic variations called single nucleotide polymorphisms (SNPs) associated with the risk of developing GDM. These SNPs not only help researchers understand the causes of GDM for potential drug development, but also have practical applications in clinical settings, which can be used to predict the risk of GDM, guide treatment choices, and manage GDM patients during pregnancy and postpartum, enhancing our understanding and approach to this condition [3].

Several genes, such as the transcription factor 7-like 2 (*TCF7L2*) gene, the haematopoietically expressed homeobox gene (*HHEX*), the adiponectin, C1Q, and collagen domain containing (*ADIPOQ*) gene, and the vascular endothelial growth factor A (*VEGFA*) gene, are expressed in major tissues contributing to GDM, including the placenta. Their SNPs have been reported to be associated with GDM. This review will explore the biological and physiological effects of these SNPs on the development of GDM. We retrieved the prediction of transcription factors on each SNP by JASPAR CORE 2022 [4] from the UCSC Genome Browser [5]. We also discuss how each tissue contributes to the pathogenesis of GDM, as shown in Figure 1.

## 2. *TCF7L2*

The transcription factor 7-like 2 gene (*TCF7L2*) is located on chromosome 10q25.3 and contains 18 exons, presenting complex splicing in different tissues. TCF7L2 is a member of the T-cell factor/lymphoid enhancer binding factor family (TCF/LEF), the transcription factor of the Wnt/β-catenin signaling pathway. Mechanistically, TCF7L2 heterodimerizes with β-catenin, then binds to DNA through a high-mobility group (HMG)-box domain to act as either a stimulator or repressor of target gene expression [6].

Several *TCF7L2* SNPs have been reported to be associated with a high risk of developing GDM in various populations [7,8,9,10]. Notably, all SNPs are in intronic regions, as shown in Table 1, suggesting the regulatory function of the SNPs in cell-context specificity. Among *TCF7L2* variants, rs7903146 (C > T) presented a strong association with GDM risk and increased more than fivefold in the TT genotype in Caucasian women [11]. In contrast, there were no significant differences in the frequencies of the *TCF7L2* rs7903146 genotypes between GDM and normal healthy women in the Chinese population; however, this SNP affected glycolipid metabolism in GDM women [9]. The risk allele at SNP rs7903146, identified through fine mapping, coincided with enhanced histone marks in both islets and adipose tissue [12]. This SNP was associated with the offspring’s birth weight [13]. Notably, *TCF7L2* SNPs rs7895340 and rs11196205 demonstrated a significant association with T2D in Thai patients similar to the association observed with SNP rs7903146 in Europeans, suggesting the effect of SNPs on the association in different ethnicities [14]. *TCF7L2* is expressed in many tissues involved in glucose and lipid metabolism, such as adipose tissue, liver, pancreas, and placenta.

Pancreas-specific and β-cell-specific *Tcf7l2* null mice altered glucose homeostasis by decreasing glucose tolerance, impairing insulin secretion, and reducing pancreatic β-cell volume [15,16]. TCF7L2 was bound to phosphoinositide-3-kinase regulatory subunit 1 (*PIK3R1*) promoter to suppress the encoding of p38, resulting in the activation of p-Akt and an increase in insulin secretion [17]. Selective deletion of *Tcf7l2* in mice pancreatic α-cells showed a lower α-cell mass and defective counterregulatory response by increasing glucose infusion rates and decreasing plasma glucose concentration. Loss of *Tcf7l2* also suppressed the expression of glucagon (*Gcg*) and MAF BZIP transcription factor B (*Mafb*) in the islet [18]. However, no effects were found on insulin (*Ins)* gene expression in insulin-positive cells and β-cell mass [18]. A *Tcf7l2* mutant in zebrafish showed hyperglycemia, pancreatic and vascular defects, and decreasing regeneration [19]. RNAseq in zebrafish adult pancreatic cell type revealed that *Tcf7l2* exhibited the highest expression in acinar tissue and was expressed in δ-cells but not in β-cells [19].

*TCF7L2* plays a role in the development and functioning of adipose tissue. Humans with impaired glucose tolerance and adipocyte insulin resistance [20], humans with obesity [12], and mice with high-fat diet-induced obesity [21] showed low *TCF7L2* levels. TCF7L2 was required for Wnt/β-catenin signaling during adipogenesis [20]. Conditional adipocyte *Tcf7l2f/f* knockout mice induced adipocyte hypertrophy and impaired lipolysis response to fasting [21]. Specifically in mature adipocytes in vivo, disabling the TCF7L2 protein by eliminating the HMG-box DNA-binding domain resulted in overall glucose intolerance and insulin resistance in the liver [20]. Transcriptome-wide profiling indicated that TCF7L2 regulates the extracellular matrix, immune response, and apoptosis in the adipose progenitor [12]. However, the actions of TCF7L2 in adipose tissue remain controversial. Human adipose progenitor had the highest expression of *TCF7L2* mRNA but then decreased during differentiation [12]. In contrast, *Tcf7l2* was elevated during adipocyte differentiation in murine 3T3-L1 cells [20]. In obesity, the protein level of TCF7L2 is reduced in whole adipose tissue but increased in adipocyte progenitor. Knockdown of *TCF7L2* in adipose progenitor showed activation of Wnt/β-catenin signaling in a dose-dependent manner, resulting in impaired proliferation and adipogenesis, while overexpression of *TCF7L2* activated adipocyte differentiation [12]. Finally, SNP rs7903146 minor allele T increased insulin accumulation in adipose tissue by decreasing the level of *TCF7L2* in adipose progenitor but not in mature adipocytes. A luciferase assay elucidated that the risk allele T abolished the enhancer activity of *TCF7L2* [12].

Mice with liver-specific *Tcf7l2* knockout (*Alb-Cre*; *Tcf7l2^f/f^*) displayed elevated fasting glucose levels without alterations in body weight, plasma insulin, or insulin-like growth factor-1 levels. The deficiency of hepatic *Tcf7l2* resulted in impaired glucose and insulin tolerance, yet no changes were observed in hepatic insulin signaling or energy expenditure. Interestingly, hepatic *Tcf7l2* depletion significantly increased the expression of lipogenic genes in the liver, without affecting genes related to β-oxidation and lipolysis [22]. 

Notably, *TCF7L2* is also exhibited in the placenta [23]; however, the mechanism associated with GDM has never been revealed.

## 3. *HHEX*

The haematopoietically expressed homeobox (*HHEX*) gene encodes a member of the homeobox family of transcription factors. The genomic structure of human *HHEX* comprises four exons. It is mapped in the 270-kb linkage disequilibrium (LD) block of 10q23.33, containing three genes: insulin degradation enzyme (*IDE*), kinesin family member 11 (*KIF11*), and *HHEX* [24]. 

*HHEX* variants rs1111875 (T > C), rs5015480 (T > C), and rs7923837 (A > G) were associated with increased susceptibility to GDM [25,26,27]. All three SNPs are in the intergenic region of the *IDE*-*KIF11*-*HHEX* LD block, in which the closest gene is *HHEX* [25]. However, in silico studies, such as cis-expression quantitative trait locus (cis-eQTL), which linked SNPs to *HHEX*, have never been confirmed. Interestingly, the major alleles in all SNPs increased the risk of GDM and are mainly predicted to be the binding sites of forkhead box (FOX) transcription factors, as shown in Table 1.

*HHEX* plays roles during pancreas and liver embryogenic development, which may affect glucose homeostasis and insulin secretion later in life. The pancreas and liver originate from a shared pool of progenitors, and HHEX plays a crucial role in development by serving as a gatekeeper for pancreatic lineage specification by collaborating with pancreatic transcription factors and endodermal transcription factors like FOXA2 and GATA4 [28]. HHEX participates in the development of the pancreatic endoderm by controlling the expression of genes associated with pancreatic development, including *NKX6.1*, *PTF1A*, *ONECUT1*, and *ONECUT3* [29]. Notably, *HHEX* was not detected in the pancreatic β-cells and α-cells of adult humans and mice, while it was exhibited in adult pancreatic δ-cells and regulated somatostatin secretion [30]. Lysine-specific demethylase-1 targeted the *Hhex* locus and promoted the repression of *Hhex* by facilitating methylation at H3K4me1/2, leading to the prevention of the transition of β-cells to δ-cells [31]. 

Although *HHEX* is expressed in adipocytes [32] and the placenta [33], studies of mechanism and function remain limited. Knockdown of *Hhex* in preadipocyte cell line 3T3-L1 suppressed adipogenesis in a dose-dependent manner and led to the suppression of peroxisome proliferator-activated receptor gamma (*PPARG*) level [32].

## 4. *SLC30A8*

The solute carrier family 30 member 8 gene (*SLC30A8*) encodes zinc transporter 8 (ZnT8) protein, which is highly specifically expressed in insulin-producing β-cells and required for insulin biosynthesis and secretion. *SLC30A8* is located on chromosome 8q24.11 and consists of eight exons spanning a length of 37 kb and encoding a 369-amino acid protein [34]. 

Three *SLC30A8* SNPs have been reported to be associated with the risk of GDM (Table 1). SNP rs13266634 (C > T) is a missense variant affecting the amino acid residue 325 in the ZnT8 C-terminal. The major allele CGG at the codon results in the production of arginine (R), whereas the minor allele TGG leads to the generation of tryptophan (W), denoted as R325W. This SNP showed significant associations with GDM and ethnicity specificity. The rs13266634 C allele increased the risk of developing GDM [35,36,37] whereas the T allele was a protective variant against the development of GDM [36,38,39]. Loss of ZnT8 function improved glucose responsiveness and enhanced proinsulin conversion, leading to more effective insulin secretion and preservation of the function of β-cells [40,41]. Mechanistically, the protective T allele enhanced the response to the glucose challenge by increasing insulin secretion and decreasing glucagon secretion in primary islets. Additionally, there was a tendency for the T allele to lower the expression of *SLC30A8* (*p* = 0.053) and genes related to proinsulin processing [42]. 

The other two *SLC30A8* SNPs, rs3802177 (G > A) and rs2466293 (A > G), are both located at 3′ UTR variants. *SLC30A8* rs3802177 [7] and rs2466293 [38] were significantly associated with an increased risk of GDM in the Chinese population and were predicted to be binding sites for transcription factors, as shown in Table 1. However, the molecular mechanisms governing how these SNPs affect the expression of *SLC30A8* require further investigation.

## 5. *ADIPOQ*

The adiponectin, C1Q, and collagen domain containing gene (*ADIPOQ*) is located on chromosome 3q27.3 and is major expressed in adipose tissue. It encodes adiponectin, a 30 kDa circulating protein in the plasma, which is involved in various physiological mechanisms encompassing energy metabolism and insulin sensitivity [43]. Women with GDM often experience hypoadiponectinemia, leading to insulin resistance and glucose intolerance [44]. 

SNPs rs17300539 (−11391G > A) and rs266729 (−11377C > G) in the promoter region, rs2241766 (+45T > G) in exon 2, and rs1501299 (+276G > T) in intron 2 have been reported as being associated with the risk of GDM development (Table 1). These four variants are situated within the two *ADIPOQ* LD blocks. Block 1 includes the promoter sequence spanning from −14,811 to −4120, while block 2 encompasses the exons within the region −450 to +4545 [45].

*ADIPOQ* rs266729 (C > G) allele G increased the risk of GDM in Asian and European populations [46,47], while reducing it in the American population [48]. The study of Chinese women did not show a significant difference in the genotypes and allele frequencies of rs266729 between GDM patients and the control [49]. The patient’s carrier allele G had a higher adiponectin level [46] and diastolic blood pressure [49] compared to carrier allele C. In contrast, the study of Bulgarian women elucidated that the G allele had a protective effect against GDM [50]. Another SNP in the promoter region, rs17300539 (G > A) allele G, was associated with lower adiponectin [51]. Notably, the variant rs17300539, along with rs266729, has been reported as having no association with GDM in black South African women [52].

SNP rs2241766 (T > G) is a synonymous mutation (Gly15Gly) at exon 2. Although rs2241766 allele G was associated with the risk of GDM [53], the effects of this SNP on adiponectin levels in serum are still controversial in different ethnicities [45,46,54,55]. In Thai women with GDM, the study showed no significant differences in adiponectin levels between different genotypes [46]. However, in Chinese patients, allele G was associated with higher adiponectin levels compared to allele T [55]. Conversely, in Malaysian patients, the trend was reversed, with allele T linked to higher adiponectin levels [54].

Rs1501299 (G > T) showed an association with the risk of T1D, T2D, and GDM [56]; however, there are still few studies on GDM [46,49,53,57]. Tangjittipokin et al. elucidated that patients with minor allele T had higher fasting glucose levels than those with major allele G [46]. GDM patients with the rs1501299 GT genotype had lower diastolic blood pressure than those with the GG genotype [49].

Taken together, the *ADIPOQ* SNPs are important susceptibility factors for developing GDM. Nevertheless, the molecular mechanisms governing how these SNPs affect the level of adiponectin need to be investigated. 

## 6. *FTO*

The human fat mass and obesity-related gene (*FTO*) is mapped on chromosome 16q12.2, including nine exons and eight introns. *FTO* encodes a 2-oxoglutarate-dependent nucleic acid demethylase, and its main substrate is N6-methyladenosine (m6A) in nuclear RNA, controlling several post-transcriptional regulatory processes [58]. *FTO* is widely expressed in adipose tissues and skeletal muscles in humans, playing a crucial role in the regulation of body weight and fat mass. *FTO* is also a critical regulator of adipogenesis that acts in the early stages of adipogenesis, leading to obesity [59]. *FTO* inhibits insulin secretion and contributes to pancreas islet β-cells dysfunction [60]. However, no study has investigated the role of skeletal muscles in the functioning of the *FTO* gene in GDM.

*FTO* rs1121980 (G > A) minor allele A was found to be significantly associated with a reduced risk of GDM [61]. Additionally, the SNP rs9939609 (T > A) showed a significant association with the risk of developing GDM, and the association was detected only in the Caucasian subgroup by meta-analysis [37]. Individuals carrying the AT genotype of rs9939609 were found to have a higher risk of exceeding excessive gestational weight gain earlier than those with the TT genotype [62]. Notably, multiple meta-analysis studies have reported an inability to detect significant associations between *FTO* polymorphisms, including rs9939609, rs8050136, rs1421085, and rs1121980, and the risk of developing GDM [63,64,65,66,67].

Although the *FTO* SNPs rs8050136, rs9939609, and rs1421085 did not emerge as major genetic regulators in the development of GDM, these SNPs were found to be associated with concentrations of adiponectin and TNFα in subjects with GDM [63]. SNP rs8050136 showed no significant association with GDM, but posed an increased risk of GDM in Bangladeshi multigravida women [67]. Interestingly, *FTO* SNPs are located in intron 1 (as listed in Table 1), suggesting a potential role in regulating the expression of *FTO* itself and possibly influencing nearby genes [58]. 

## 7. *VEGFA*

The vascular endothelial growth factor A (*VEGFA*) gene is located at chromosome 6p21.3 and consists of eight exons separated by seven introns with a full molecular length of 14 kb. *VEGFA* undergoes alternative mRNA splicing, resulting in the generation of different-length isoforms. These isoforms possess distinct biological properties, and each plays a specific role in the differentiation and development of the vascular system [68]. 

Hypervascularity is normally found with increase in placental weight in GDM [69]. However, the plasma levels of VEGFA are still contradictory. Troncoso et al. showed no difference in VEGFA levels between GMD and the control group, while Meng et al. indicated that GDM patients had lower VEGFA levels [70]. In contrast, other studies showed that GDM patients had elevated VEGFA levels [71,72]. *VEGFA* is also expressed in β-cells and controls development and function. The protein level of VEGFA remained constant in β-cells throughout pregnancy in mice [73]. However, when VEGFA signaling was disrupted in β-cells of pregnant mice, it led to a decrease in the maintenance of islet vessels and induced transient glucose intolerance without affecting the expansion of the β-cell mass [74]. 

Although the expression of *VEGFA* seems important in the pathogenesis of GDM, there is only one study on the association of *VEFGA* polymorphisms with GDM in Chinese women [71]. Five SNPs of the *VEGFA* gene were selected: rs2146323 (C > A), rs2010963 (G > C), rs3025039 (C > T), rs3025010 (T > C), and rs833069 (G > A) (Table 1). SNPs rs2146323 allele A (intron 2) and rs3025039 allele T (exon 7) showed a dramatically increased risk of GDM. The patients who carried the rs3025039 CT+TT genotype had a higher level of VEGFA than those who carried the CC genotype. Other SNPs showed no difference in the distributions of genotypes and alleles between GDM and control. Interestingly, the frequency of haplotypes CAAC, CAAT, CACC, CACT, GACT, and GGCT for the rs2010963–rs833069–rs2146323–rs3025010 combination was found to be statistically different between women with GDM and healthy women [71]. 

An imbalance in the expression of various other vascular endothelial growth factors (*VEGFs*) and their receptors (*VEGFRs*) also influences the pathophysiology of the GDM [75], future studies examining susceptibility variants and the underlying biological mechanisms will likely unveil associations within this growth factor family with GDM.

## 8. *CDKAL1*

The CDK5 regulatory subunit associated protein 1-like 1 gene (*CDKAL1*) is in chromosome 6p22.3. This gene encodes a 65-kDas CDKAL1 protein, functioning as a methylthiotransferase. It acts as a tRNA modification enzyme, playing a role in enhancing the translation of various transcripts in pancreatic β-cells, including that of proinsulin. CDKAL1 also serves as an endogenous inhibitor of cyclin-dependent kinase 5 (CDK5). The inhibition of CDK5 by CDKAL1 plays a crucial role in preventing the translocation of PDX1, a transcription factor responsible for regulating the insulin gene. This disruption in the translocation process leads to an interruption in insulin production [76]. In adipocytes, CDKAL1 negatively regulates the adipocyte-specific genes in response to hyperglycemic and hyperlipidemic conditions [77]. CDKAL1 regulates the differentiation of adipocytes in murine 3T3-L1 cells through the Wnt/β-catenin signaling pathway [78]. Additionally, in adipose-specific *Cdkal1* knockout mice, CDKAL1 is involved in controlling mitochondrial function [79].

The GDM-associated SNPs fall within an intronic 5 region of the *CDKAL1* locus (Table 1) and have recently been the hot spot for studying the risk with GDM. SNP rs7754840 (G > C) allele C significantly correlated with GDM risk in meta-analysis [64,80,81]; however, the associations varied across different ethnicities. No association between variations and GDM was found in Egyptian and Chinese populations [82,83]. Meanwhile, Caucasian women showed an association, and Bangladeshi women had a significant association only after adjusting for gravidity and family history of diabetes [39,84]. Meta-analysis showed that SNP rs7756992 (A > G) minor allele G increased the risk of GDM in Bangladeshi women [64,80,84]. The allele G of this SNP also showed a strong association with impaired glucose metabolism, low birth weight, and a decreased insulin secretion index later in life [85]. Interestingly, SNP rs7747752 (G > C) was reported as being associated with GDM risk without the previous report on association with T2D. The studies conducted on Chinese women elucidated that SNP rs7747752 allele C was genetically associated with elevated GDM risk and with interaction with other factors [86,87,88]. Another SNP, rs9368222 minor allele A, which is in the same block as rs7747752, was also associated with the risk of GDM in the Hispanic population but not in the Caucasian population [89]. For SNP rs10946398 (A > C), well known to be associated with T2D, it was recently reported that allele C was associated with GDM in Pakistani women [90]. In contrast, the CC genotype was not associated with GDM, but was associated with the need for insulin therapy in Caucasian women [91]. Notably, the *CDKAL1* locus stands out as a prominent example of ancestry-correlated heterogeneity, as multi-ancestry meta-analyses showed that the SNPs at this locus exhibited more pronounced effects on GDM in GWAS conducted on individuals of East Asian ancestry compared to other populations [92].

## 9. *MTNR1B*

The melatonin receptor 1B gene (*MTNR1B*), located in chromosome 11q14.3, encodes melatonin receptor MT2, which is expressed in the human brain, pancreatic β-cells, and skeletal muscle. MT2 not only regulates circadian rhythm, but is also involved in glucose metabolism. Melatonin binds with MT2 and suppresses cyclic guanosine monophosphate (cGMP) signaling, which results in a decrease in insulin secretion in pancreatic β-cells. In the skeletal muscle, the binding of melatonin to MT2 stimulates glucose transport into cells via the insulin receptor substrate-1 (IRS-1)/phosphoinositide 3-kinase (PI-3-kinase) pathway [93]. *MTNR1B* is also expressed in the placenta, and GDM women have higher levels of MT2 compared to normal women [94]. A study of trophoblast HTR-8/SVneo cells showed that melatonin upregulates GLU4, PPARG, and MT2 expression, leading to elevated glucose consumption [94].

Several *MTNR1B* polymorphisms were shown to be associated with GDM (Table 1). SNP rs10830963 (C > G), within the single 11.5-kb intron, is the most popular variant, and allele G showed susceptibility to increased risk of GDM from meta-analysis and different ethnicities [7,13,35,39,66,89,94,95,96,97]. The placental expression level of MT2 in GDM women carrying the GG and GC genotypes was higher than in those carrying the CC genotype [95]. SNPs rs1387153 (C > T) allele T and rs10830962 (C > G) allele G are both located in the intergenic region over 2 kb upstream from *MTNR1B* and demonstrated a significant association with an increased risk of GDM [96,98,99,100,101]. Notably, maternal rs10830963 (C > G) had an association with offspring birth weight [13,102]. SNP rs4753426 (T > C), located in the *MTNR1B* promoter, still has contradictory reports. No statistically significant differences were observed in the distribution of *MTNR1B* rs4753426 genotypes and alleles between women with GDM and healthy pregnant women in a study conducted on the Caucasian population [66]. However, rs4753426 minor allele C was associated with susceptibility to GDM [96,98], and another study demonstrated that major allele T was a protective variant for GDM development [99].

## 10. *GLO1*

The glyoxalase I gene (*GLO1*) is located on chromosome 6p21.2 and is predominantly expressed in the human skeleton. GLO1 serves as a protective enzyme against dicarbonyl stress. Mechanistically, methylglyoxal (MG) is a byproduct of glycolysis, and GLO1 plays a crucial role in detoxifying MG to D-lactate. Elevated levels of MG can directly inhibit insulin signaling by binding to IRS-1 within the skeletal muscle and by binding to circulating insulin. Conditions associated with metabolic impairment, such as obesity, insulin resistance, and T2D, have been observed to suppress the expression of *GLO1*, disrupting the MG/GLO1 axis in skeletal muscle metabolism [103].

The association between *GLO1* polymorphisms and GDM has only one report from Zeng et al., who recently showed the effects in the Chinese population [104]. SNP rs1130534 (T > A) is a synonymous mutation at codon 124 (Gly124Gly). The AA genotype of rs1130534 was identified as a protective factor against GDM. The TA genotype increased fasting glucose levels and the risk of GDM. Furthermore, newborn weight was notably lower in individuals with the TA genotype than in those with the TT genotype. SNP rs4746 (T > G) is a missense variant affecting the amino acid residue 111 in exon 4. The codon major allele GAG generates a glutamic acid (E), while the minor allele GCG instead generates an alanine (A) (E111A). The GG genotype and the G allele were associated with a lower risk of GDM [104]. Another *GLO1* variant, rs1781735, is in the promoter, which significantly affects the transcriptional activity of *GLO1* [105]. This SNP showed no association with the risk of GDM, but the GG genotype was more associated with cumulative neonatal weight and MG levels than the TG or TT genotypes. Also, individuals with the GG genotype exhibited significantly higher MG levels than individuals with the TT genotype. The combined effects of multiple SNPs, specifically the rs1781735–rs4746–rs1130534 (T–G–T) haplotypes, were found to significantly reduce the risk of developing GDM [104]. Notably, none of the *GLO1* variants is predicted to be the binding site of the transcription factor, as shown in Table 1.

## 11. *GCK* and *GCKR*

The glucokinase gene (*GCK*) is situated on chromosome 7p15.3, composed of 10 coding exons, and is expressed in pancreatic β-cells, liver, and brain. On the other hand, the glucokinase regulator gene (*GCKR*) is located on chromosome 2p23.3 and encodes a glucokinase regulatory protein (GKRP), responsible for controlling the activity of GCK. GCK, in turn, plays a key role in glucose storage and disposal in the liver and modulates insulin secretion in the pancreas. Mechanistically, GKRP forms an inactive complex with GCK at basal glucose concentrations. Together, GCK and GKRP work in tandem, maintaining blood glucose homeostasis. Variants in the *GCK* and *GCKR* genes can disrupt the balance of the GCK/GKRP complex, leading to abnormal glucose concentrations and hyperglycemia [106,107]. Indeed, heterozygous inactivating mutations in *GCK* can lead to a form of diabetes known as maturity-onset diabetes of the young (GCK-MODY). This condition presents a challenge in diagnosis, especially during pregnancy, as it can easily be confused with GDM [108]. 

*GCK* SNPs rs1799884 (C > T) and rs4607517 (G > A) were both reported to be associated with T2D; however, the association was different in GDM. SNP rs1799884 (C > T) is in the *GCK* promoter without any study and prediction to be the binding site of transcription factor, as shown in Table 1. The association of this variant with GDM is still contradicted in Asian and Caucasian populations. *GCK* rs179988 minor allele T was significantly associated with GDM [39,109,110] but no association was reported [61,86,91]. GDM women showed a predominance of the T allele and more commonly required insulin treatment [91]. SNP rs4607517 (G > A), an intergenic variant between *GCK* and *YKT6*, had no report to show an association with GDM [61,86,109], except for the interaction between this *SNP* and sweets consumption on GDM [111]. However, fasting blood glucose was significantly lower in individuals with the rs4607517 GA genotype than in those with the GG genotype [109]. Notably, the distribution of rs4607517 was not in Hardy–Weinberg equilibrium in the Filipinos [35]. 

SNP rs1260326 (C > T) is a missense variant at *GCKR* exon 15, which substitutes proline to leucine at position 446 (P446L). Nevertheless, there was no difference in genotypes between cases and controls [11] and no association of rs1260326 with GDM risk was found from the meta-analysis [37,64]. Interestingly, the association between rs1260326 major allele C and GDM was shown in Chinese and Euro-Brazilian women [49,112], among whom GDM women with the CC genotype had a higher 1-h oral glucose tolerance test (OGTT) level compared to those with the TT genotype [49]. In contrast, the *GCKR* intronic rs780094 (T > C) variant was the risk factor for GDM from the meta-analysis [37,64]. However, the case-control studies showed no difference in genotypes between control and GDM women [91], and no association of the SNP with GDM risk was observed [109].

## 12. Conclusions

The potential to comprehend the pathophysiology of GMD through genetic polymorphisms (Figure 2) holds promise, providing insights into predicting and managing GDM patients’ risks. However, the incorporation of associated SNPs into clinical practice remains a challenge, influenced by such factors as GDM detection criteria, sample size, and ethnicity in GWAS analysis. Additionally, the shared pathogenesis with T2D complicates the identification of unique GDM variants. Given the multicomplex nature of GDM involving various tissues, it is crucial to uncover the underlying mechanism of each genetic variant. Furthermore, genetic information must be considered alongside factors such as diet, lifestyle, and gut microbes that influence individual susceptibility. The continuous exploration of new SNPs, improving the genetic background of GDM, and integrating machine learning into polygenic risk score (PRS) models may have the potential to refine our understanding and offer valuable assistance in clinical practice.

## Figures and Tables

**Figure 1 ijms-25-02039-f001:**
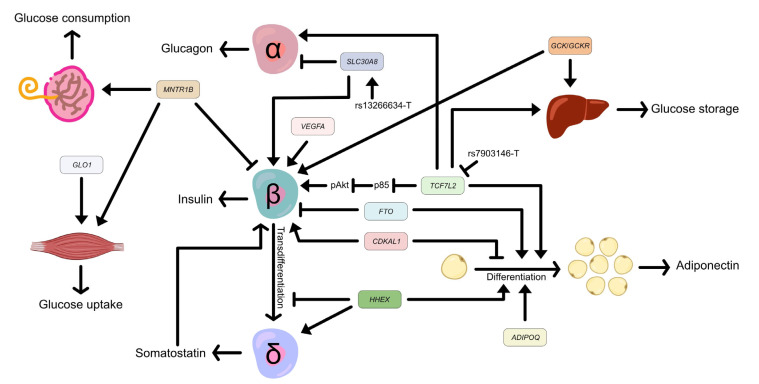
Effects of associated genes and SNPs in major tissues contribute to gestational diabetes mellitus. *TCF7L2* promotes glucose storage in the liver, activates insulin secretion in β-cells by suppressing the expression of p85, and inhibits adipogenesis. The *TCFL2* rs7903146 minor allele, T, downregulates the expression of *TCF7L2*. *HHEX* increases somatostatin secretion and adipocyte differentiation but inhibits the transdifferentiation of β-cells to δ-cells. *SLC30A8* activates insulin secretion and reduces glucagon secretion. The *SLC30A8* missense variant rs13266634 allele T improves glucose responsiveness and enhances proinsulin conversion. *ADIPOQ* activates adipogenesis. *FTO* inhibits insulin secretion and promotes adipocyte differentiation, and vice versa by *CDKAL1*. *VEGFA* maintains β-cell function. *MTNR1B* increases glucose consumption in the placenta and skeletal muscle, but decreases insulin secretion in pancreatic β-cells. *GLO1* activates glucose uptake in skeletal muscle and *GCK*/*GCKR* promotes glucose storage in the liver. The transcription factor 7-like 2 (*TCF7L2*) gene, the haematopoietically expressed homeobox (*HHEX*) gene, the solute carrier family 30 member 8 gene (*SLC30A8*), the adiponectin, C1Q, and collagen domain containing (*ADIPOQ*) gene, the human fat mass and obesity-related (*FTO*) gene, the CDK5 regulatory subunit associated protein 1 like 1 (*CDKAL1*) gene, the vascular endothelial growth factor A (*VEGFA*) gene, the melatonin receptor 1B (*MTNR1B*) gene, the glyoxalase I (*GLO1*) gene, the glucokinase (*GCK*) gene, and the glucokinase regulator (*GCKR*) gene.

**Figure 2 ijms-25-02039-f002:**
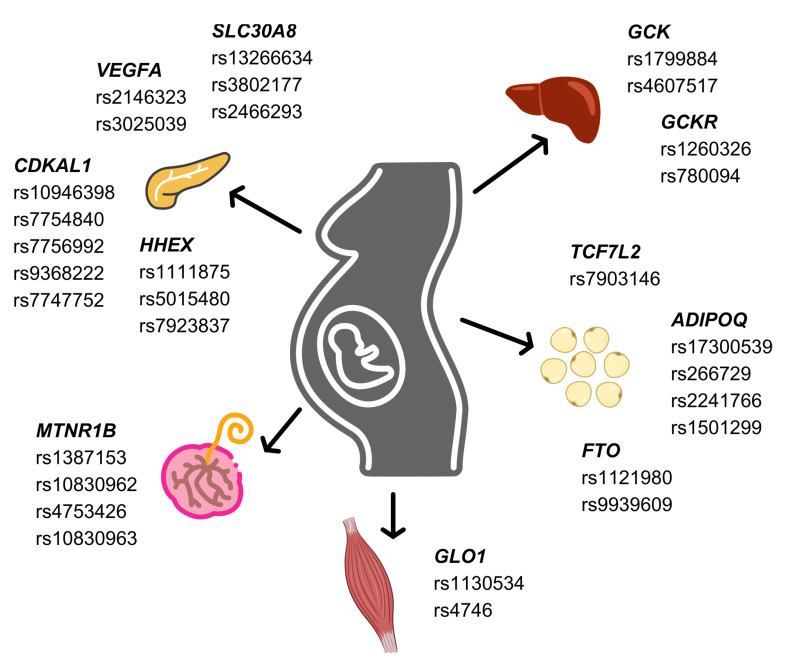
Associated single nucleotide polymorphisms on gestational diabetes mellitus risk in pancreas, liver, skeletal muscle, adipose tissue, and placenta.

**Table 1 ijms-25-02039-t001:** Gestational diabetes mellitus–associated SNPs.

Genes	SNPs	Location	Risk Allele	Predicted TF (JASPAR)	Expression in Placenta
*TCF7L2*	rs34872471	Intron 4	C	ZNF354A, Six3, JUND	Y
	rs7901695	Intron 4	C	n/a	
	rs4506565	Intron 4	T	PLAGL2	
	rs7903146	Intron 4	T	ZNF211	
	rs12243326	Intron 4	C	ZNF211	
	rs12255372	Intron 5	T	n/a	
	rs290487	Intron 8	T	NFIC::TLX1	
*HHEX*	rs1111875	Intergenic	C	ETV2::FOXI1, ERF::FOXI1, FOXO1::FLI1, SPIB, GABPA, Erg	
	rs5015480	Intergenic	C	n/a	Y
	rs7923837	Intergenic	G	FOXO, FOXA, FOXP	
*SLC30A8*	rs13266634	Missense	C	PAX1, PAX2, PAX9	N
	rs3802177	3′ UTR	G	PAX1, PAX9	
	rs2466293	3′ UTR	G	NR1D2, RXRA::VDR	
*ADIPOQ*	rs17300539	Promoter	A	n/a	
	rs266729	Promoter	G	THAP1	
	rs2241766	Exon 2	G	Tfcp2l1, RARB, RARG	
	rs1501299	Intron 2	T	ZNF354A, POU4F3	
*FTO*	rs1421085	Intron 2	No association	ONECUT1, ONECUT3, CUX1, CUX2	Y
	rs1121980	Intron 2	G	SIX2	
	rs8050136	Intron 2	No association	ONECUT1, ONECUT2, CUX1	
	rs9939609	Intron 2	A	POU2F1::SOX2	
*VEGFA*	rs2010963	Exon 1	No association	ZNF701, IRF2	Y
	rs833069	Intron 2	No association	ZNF257, ZKSCAN5, ZNF701, ZNF263, SP4, SP5	
	rs2146323	Intron 2	A	Zfp335	
	rs3025010	Intron5	No association	Ebf2, EBF3, ZNF449, ZNF682, KLF3, NFIC, ZNF324	
	rs3025039	Exon 7	T	Bach1::Mafk, Mafg, MAFG::NFE2L, MAFK, MAF::NFE2	
*CDKAL1*	rs10946398	Intron 5	C	CREB1, MAF	Y
	rs7754840	Intron 5	C	n/a	
	rs7756992	Intron 5	G	Irf1, IRF7, MEF2A, MEF2C, MEF2D	
	rs9368222	Intron 5	A	SOX4, SOX10, SOX 12, Sox 11, Sox6	
	rs7747752	Intron 5	C	MGA	
*MTNR1B*	rs1387153	Intergenic	T	NR5A1, ZBTB12, ZSCAN31, THAP1	Y
	rs10830962	Intergenic	G	MEF2B, MEF2D, Rhox11, CDX2	
	rs4753426	Promoter	C	n/a	
	rs10830963	Intron 1	G	n/a	
*GLO1*	rs1130534	Exon 4	T	n/a	Y
	rs4746	Exon 4	T	n/a	
	rs1781735	Promoter	No association	n/a	
*GCK*	rs1799884	Promoter	T	n/a	Y
	rs4607517	Intergenic	A	PBX3, SREBF1, PKNOX1, Neurod2	
*GCKR*	rs1260326	Missense	C	n/a	Y
	rs780094	Intron 16	C	NRL, TBX19, TBXT, PKNOX1

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
