# Peer review of "Mechanisms and Physiological Roles of Polymorphisms in Gestational Diabetes Mellitus"

_ijms, 2024, doi:10.3390/ijms25042039_

Round 1

Reviewer 1 Report

Comments and Suggestions for Authors

This is an interesting review of 11 genes/loci implicated in gestational diabetes.  This is an narrative review, which is researched and written well with appropriate referencing. Authors have reviewed these genes and have assessed the prediction of transcription factors on each gene using JASPAR.  The review is comprehensive and up to date.  Each SNP/gene is discussed with reference to physiological/biological functions and genetic variation with implication for personalized medicine and patient stratification.  The review is well written and but figures 1& 2 could be high resolution

Reviewer 2 Report

Comments and Suggestions for Authors

The present article entitled “Mechanisms and Physiological Roles of Polymorphisms in 2 Gestational Diabetes Mellitus” drafting, designing and presentation are very well organized. But there is some missing information in introduction and conclusion section to be include in the revision of the current manuscript.  Many typo and grammatical errors also found in the text. Therefore, I requested to authors for minor revision of the current manuscript

Comments on the Quality of English Language

There are some typo and grammatical error in the text to be removed in revised manuscript.

Reviewer 3 Report

Comments and Suggestions for Authors

The authors retrieved predicted transcription factors on SNPs of GDM-associated genes and compiled the literature related to the genes. Thorough review of literature and linking the genes to functions of organs associated with glycemic control as in Fig. 1. are appreciable.

Line 46: “This review will concentrate on the cellular signaling and molecular mechanisms involved in the development of GDM” is misleading. The review focuses only on “the SNPs associated with this condition”

Needs proof reading: e.g. line 53: liver, not live.

Names of genes need italicization throughout the document. Please follow the guidelines on formatting the name of genes and proteins.

Fig. 1’s title needs to be changed. “Pathophysiological mechanisms in major tissues contribute to gestational diabetes mellitus” is misleading as the figure is mainly on the genes associated with GDM.

Comments on the Quality of English Language

Needs proof reading (e.g. line 53: liver, not live).

Names of genes need italicization throughout the document. Please follow the guidelines on formatting the name of genes and proteins.
